# A data exploration tool for averaging and accessing large data sets of snow stratigraphy profiles useful for avalanche forecasting

Florian Herla[1], Pascal Haegeli[1], and Patrick Mair[2]

[1]Simon Fraser University, Burnaby, BC, Canada
[2]Harvard University, Cambridge, MA, USA

**Correspondence:** Florian Herla (fherla@sfu.ca)

**Abstract.** Snowpack models can provide detailed insight about the evolution of the snow stratigraphy in a way that is not possible with direct observations. However, the lack of suitable data aggregation methods currently prevents the effective use of the available information, which is commonly reduced to bulk properties and summary statistics of the entire snow column or individual grid cells. This is only of limited value for operational avalanche forecasting and has substantially hampered the application of spatially distributed simulations as well as the development of comprehensive ensemble systems. To address this challenge, we present an averaging algorithm for snow profiles that effectively synthesizes large numbers of snow profiles into a meaningful overall perspective of the existing conditions. Notably, the algorithm enables compiling of informative summary statistics and distributions of snowpack layers, which creates new opportunities for presenting and analyzing distributed and ensemble snowpack simulations.

## 1 Introduction

The layered nature of the snowpack is a necessary condition for the formation of snow avalanches (e.g., Schweizer et al., 2003, 2016; Reuter and Schweizer, 2018), and information about the snow stratigraphy is crucial for developing a meaningful understanding of existing avalanche conditions (Statham et al., 2018). Snowpacks are inherently spatially variable due to the complex interactions of the meteorological forcing and terrain (Schweizer et al., 2007), and layer depths, thicknesses and properties can therefore vary substantially between different locations even over short distances. In some circumstances, some layer sequences might even be missing entirely. To understand the conditions at various spatial scales, avalanche forecasters observe snow profiles at targeted point locations, and then synthesize the gathered information into a mental model of the regional scale snowpack conditions, which are often represented in hand-drawn summary profiles. The documented layers in these idealized snow profiles represent key features of the conditions that forecasters expect to exist within their region. Local field observations are then used to validate and localize the regional understanding of the conditions. As the season progresses, forecasters continuously revise their mental model and update their summary profile throughout the winter as new observations become available.

While avalanche forecasters have developed meaningful strategies for synthesizing limited numbers of manual snowpack observations, the potential volume of data generated by snowpack simulations is too vast for human processing (Morin et al.,

2020). While effective visualization designs can help guide human perception to data features that prompt human reasoning (Horton et al., 2020b), visualizations of large data sets that include both spatial and temporal dimensions remain challenging. Since computer-based tools excel at applying repetitive tasks to big data sets, numerical data aggregation algorithms have the potential to allow avalanche forecasters to make better use of large scale snowpack simulations. Inspired by Hagenmuller and Pilloix (2016), Herla et al. (2021) developed a set of numerical algorithms for comparing multidimensional, mixed data type

snow profiles based on *Dynamic Time Warping*, a well established algorithm for measuring similarity between two potentially misaligned sequences. However, the medoid approach (Herla et al., 2021) employed for computing representative profiles has substantial limitations. Since the medoid is simply the profile within a given group that is most similar to all other profiles, it does not actually aggregate the available information and therefore does not necessarily represent the snowpack features that exist within the entire group meaningfully. Furthermore, it is not suited for tracking average conditions over time as the medoid

within a group of profiles can differ between time steps resulting in a disjointed and difficult to interpret time series. Finally, medoid calculations are computationally costly and thus only of limited applicability in operational contexts. All these reasons make the medoid aggregation approach unsuitable for avalanche forecasting.

The objective of this contribution is to introduce an averaging algorithm for snow profiles that extends the snow profile processing tools of Herla et al. (2021) with a global averaging method that is based on the approach proposed by Petitjean

et al. (2011). Our goal is to compute an average snow profile that provides a quick and familiar overview of the predominant snowpack features that are captured within large sets of profiles in a way that is informative for operational avalanche forecasting and supports existing assessment practices. To do so, our averaging approach needs to highlight critical snowpack features and facilitate their averaging and tracking over space and/or time. Furthermore, our approach needs to offer simple access to distributions of layer characteristics from simulated profiles to provide useful insight about the nature of the conditions. Ana-

lyzing large volumes of snowpack simulations in these novel ways will make it much easier for users to access data features and create data views relevant for avalanche forecasting. The algorithm described in this paper has been implemented in the open source R package `sarp.snowprofile.alignment` (Herla et al., 2022a), and is freely available to researchers and practitioners.

## 2 Description of the snow profile averaging algorithm

Our approach is based on Petitjean et al. (2011) who developed an averaging method specifically for sequential data called *Dynamic Time Warping Barycenter Averaging* (DBA), which builds on the comparison method for sequential data called Dynamic Time Warping. Unlike the medoid aggregation approach used by Herla et al. (2021), the average sequence derived with DBA consists of an entirely new sequence that represents the notion of an average of all individual sequences. This makes this approach more suitable for snow profile applications because it actually provides an average perspective of the conditions.

In addition, DBA also uses considerably less computation time since DBA does not rely on pairwise comparisons across the entire data set like the medoid computation, but rather uses a simple yet clever trick to apply Dynamic Time Warping not

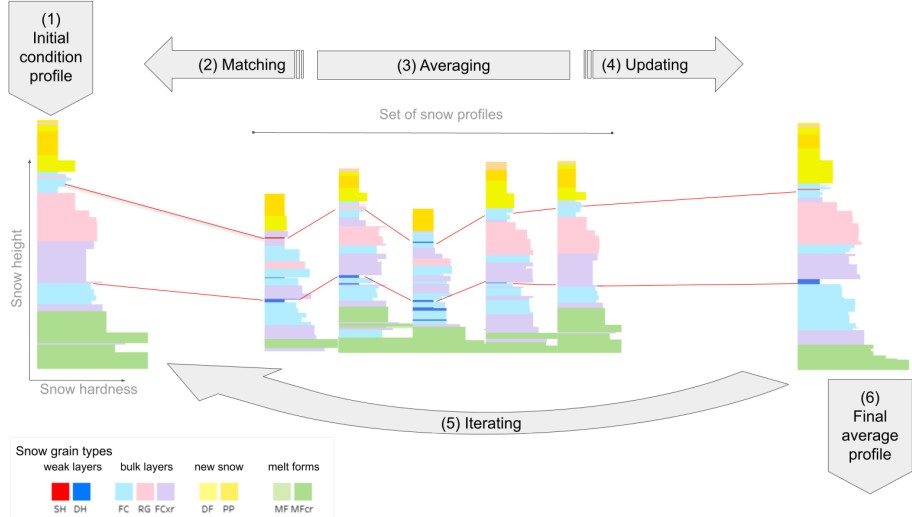

**Figure 1.** Conceptual flowchart demonstrating how Dynamic Time Warping Barycenter Averaging (DBA) can be applied to snow stratigraphy profiles. (1) An initial condition profile is picked from the data set of profiles to act as a reference profile. (2) All layers from all profiles in the set are matched against the reference profile (red line segments highlight two sets of corresponding layers), and (3) all sets of corresponding layers are averaged to (4) update the reference profile. (5) This process is repeated several times until the reference profile does not change substantially anymore and therefore (6) represents the final average profile of the set.

only to pairs of sequences but to many sequences. This is accomplished by comparing all individual sequences to a reference sequence and repeating that step in an iterative process.

The general workflow of our DBA implementation for snow profiles starts by picking an initial condition profile that acts

as the reference (Figure 1). Our Dynamic Time Warping implementation for snow profiles (Herla et al., 2021) is then used to align all individual profiles in the data set to the reference. By matching all layers from all individual profiles against all layers in the reference profile, this step creates sets of corresponding layers. All sets of corresponding layers are then averaged, and the averaged layer properties from each set are used to update the corresponding reference layers. This process of matching and updating the reference profile is repeated several times until the reference profile does not change substantially anymore. At

this point, the reference profile represents the average profile of the data set. Figure 1 illustrates the workflow of the algorithm based on a small set of profiles. The red lines highlight how two sets of corresponding layers in the individual profiles are matched to two layers in the reference profile. Due to the updating of the reference layers, the updated reference profile on the right shows different layer properties than the original reference profile on the left.

The snow profile alignment algorithm that matches each individual profile against the reference is implemented as described

in Herla et al. (2021) with one exception. While the original approach for the alignment required all snow profiles to be rescaled to identical snow heights before their alignment, this requirement has been removed for the updated implementation presented

here. All profiles can therefore be aligned on their native height grids. Since there are meaningful use cases for both approaches, the updated version of our R package allows users to choose between the two options. We have found alignments on the native height grid to be of slightly higher quality than alignments of rescaled profiles, but more importantly, alignments on native height grids are easier to interpret.

Following the approach of Huang (1998) for summarizing categorical variables in k-means clustering, we average sets of corresponding layers by first calculating the predominant *grain type* (i.e., the grain type mode). We start the averaging of mixed data type snow stratigraphy layers like this because grain type is a fundamental layer characteristic and plays a critical role in snow profile processing tools described by Herla et al. (2021). The average of other (ordinal or numerical) layer properties are then expressed by the median properties of the layers of the predominant grain type.

Since thin weak layers are particularly important for avalanche forecasting, our algorithm includes a flexible approach for ensuring critical weak layers are meaningfully included in the averaged profiles even if their grain types are not necessarily consistent and the most prevalent. Instead of averaging weak layers solely based on their grain type mode, we implemented a subroutine that can be used to label *layers of interest* based on users' particular needs and data availability. While layers of interest are typically weak layers, they can also be other layers such as crusts. These layer labels can be based on grain type classes (e.g., all persistent grain types), or they can include additional relevant measures like stability thresholds (e.g., threshold sums, SK38, p_unstable, etc.; Schweizer and Jamieson, 2007; Monti and Schweizer, 2013; Monti et al., 2016; Mayer et al., 2022). If the majority of corresponding layers is labeled as layers of interest, the resulting averaged layer properties are the median properties of all labeled layers regardless of the actual grain types. Note that this approach still eliminates weak layers that only occur in a few profiles but might still be relevant for avalanche forecasting. To address this issue, users can either query the profile set for the list of weak layers that are not included in the average profile, or they can change the hyperparameter that specifies the required occurrence frequency threshold for labeled layers to be included in the average profile away from the default 50 %. This ensures that the final average profile represents the predominant and/or relevant snowpack features and that layer properties are internally consistent.

While the stochastic and iterative nature of the DBA approach is responsible for its computational efficiency[1], it also makes it sensitive to initial conditions. We turned this potential weakness into an opportunity to steer the averaging algorithm in a more informative direction by making the algorithm choose several different initial condition profiles strategically.

Since it is important that relevant thin weak layers are represented in the average profile, we designed the following selection routine for initial conditions. The profiles-to-be-averaged are organized into several tiers based on the total number of layers of interest and the number of depth ranges [2] occupied by at least one layer of interest. Tier 1 contains all profiles with the maximum total number of layers of interest and the maximum number of occupied depth ranges. Tier 2 consist of the remaining profiles with the maximum number of occupied depth ranges and an above-average number of layers of interest, and tier 3 includes all remaining profiles with fewer occupied depth ranges but still an above-average number of layers of interest. Depending on

---

[1] While computing the medoid for a given profile set of length $N$ requires $\mathcal{O}(N^2)$ profile alignments, the DBA approach requires $\mathcal{O}(N \cdot I \cdot IC)$, where $I$ is the number of iterations and $IC$ is the number of different initial conditions.

[2] The default depth ranges are [0, 30), [30, 80), [80, 150), [150, Inf) (cm), but can be modified by the user if necessary.

how many initial conditions are requested by the user, the algorithm picks profiles from the three tiers in descending order.

While this approach ensures that the algorithm picks appropriate starting conditions by itself, the user can still customize this process by labeling relevant layers of interest (see previous paragraph). The described strategic selection of initial conditions makes prevalent weak layers more likely to get matched and included in the final average profile. Weak layers that exist in the initial condition profile but not in the rest of the data set are automatically averaged out during the first iteration.

In addition to the strategic selection of the initial condition profile, we rescale its depth to the median snow depth to maximize

the potential for meaningful layer matches and ensure that the final profile represents the snow depth distribution of the data set in a meaningful way. To avoid exaggerated rescaling, we only select initial condition profiles whose total snow depth is within the interquartile range of the snow depth distribution. More details on the actual influence of the initial condition on the final result are presented in Section 4.2.

After several DBA iterations the reference profile will only change marginally. To assess the iterative changes between the

115 reference profiles and stop the iteration cycle, we use a similarity measure for snow profiles analogous to Herla et al. (2021). The algorithm is stopped when the similarity between the reference profiles of two subsequent iterations is beyond a certain threshold. Reaching a similarity threshold of 0.99 usually takes fewer than five iterations. However, if computational speed is of the essence, using a threshold of 0.90 that is attained in two consecutive iterations yields comparable results. If the algorithm is started with several initial conditions, the best average profile among the different realizations is chosen by converting the

120 similarity measure between the reference profile and the profile set to a root mean squared error (RMSE). The average profile with the lowest RMSE is chosen as the final realization.

## 3 Applications

In this section we present several application examples to illustrate the capabilities of our algorithm. While the snow profile data set used in these examples was simulated with the Canadian weather and snowpack model chain (Morin et al., 2020) with

125 the goal to represent flat-field conditions, our tool can be applied to any simulated snow profile irrespective of its source model. Furthermore, it is possible to use our algorithm on manual profiles, but the processing of these data sets has some unique challenges (see limitation section for more details).

### 3.1 An inconspicuous asset: Overview first, details on demand

Figure 2 illustrates how a calculated average snow profile can efficiently synthesize a large volume of snow profile data into

130 an overall perspective of existing conditions in a meaningful way. Panel a of the figure shows the individual simulated profiles from 112 model grid points within a forecast region from the same day ordered by snow depth, and Panel c shows the average profile that summarizes the characteristics of the entire profile set. The average profile highlights three distinct weak layers buried in the mid to lower snowpack with a thick and consolidated slab above. Close to the surface, the snow is loose, and a surface hoar layer (SH) is starting to get buried. This overview provides an insightful synopsis and important context for

interpreting the layer sequences of individual profiles if more specific information is required.

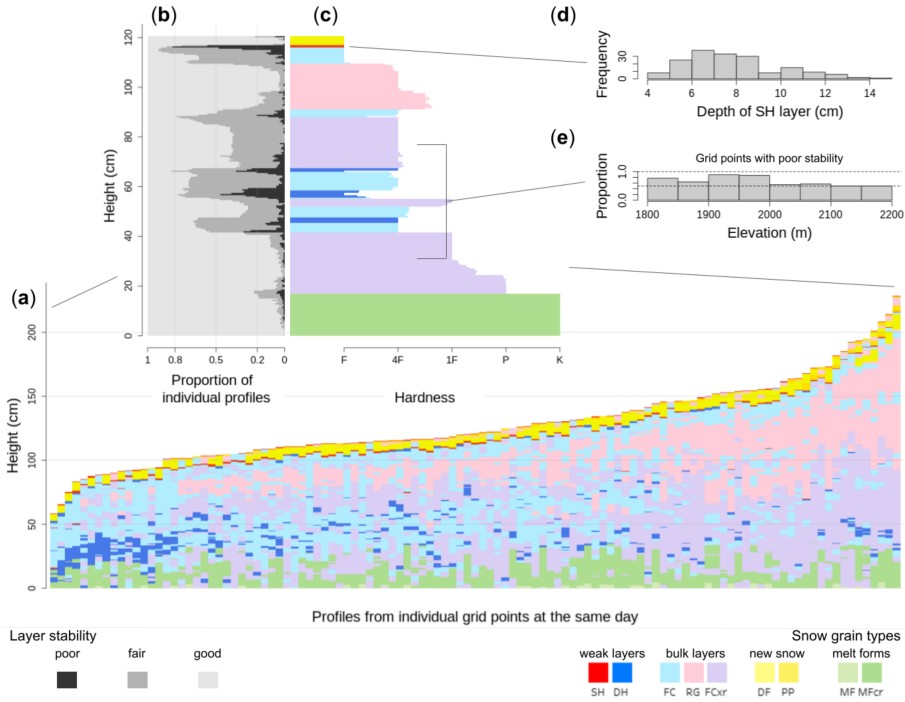

**Figure 2.** (a) Snow profile set from an avalanche forecast region, (c) which is summarized by the average profile. The average profile provides access to distributions of layer and profile properties, such as e.g. (b) the distribution of layer stabilities derived from threshold sums, (d) the depth distribution of a SH layer that is starting to get buried, (e) the elevation distribution of the proportion of profiles that contain layers with poor stability in mid snowpack.

However, there is more to the averaging algorithm than just providing a graphic representation of the average condition across a forecast area. Since all individual layers are matched against the average profile, each of the averaged layers can be traced throughout the entire data set. The average profile therefore acts as a navigation tool that connects all layers and enables the tracking of layers across space as well as the computation of distributions of layer and profile properties. Hence, the average profile embodies the important and broadly used "overview first, details on demand" data visualization principle that was first proposed by Shneiderman (1996). To illustrate this capability, Panel b in Fig. 2 shows the stability distribution of each layer using threshold sums TSA (also known as lemons) (Schweizer and Jamieson, 2007; Monti et al., 2012, 2014), which we classified into three categories (poor: $\geq 5$; fair: $\geq 4$; good: $< 4$). About 50 % of all profiles in the data set exhibit structural instability on a predominant depth hoar layer (DH) in mid snowpack, and almost all profiles contain a surface hoar layer with poor stability that will likely become a concern when buried more deeply. To dig deeper, it is easy to retrieve the depth distribution of that shallow surface hoar layer and confirm that its burial is generally quite shallow (burial depth mode: 7 cm), but there are locations with deeper burials of up to 15 cm of new snow (Panel d in Fig. 2). Note that similar charts can be computed for other stability indices or any other layer properties available in the profiles, and that calculating

distributions on subsets of layers with particular properties is also straightforward. For example, Panel e in Fig. 2 shows the elevation distribution of the proportion of profiles that contain layers with poor stability in mid snowpack. While the individual profiles shown in Panel a of the figure suggest that these weak layers mainly exist in shallower profiles, the bar chart shown in Panel e further highlights that these layers are more likely to be found at lower elevations. In summary, the average profile enables efficient and user-friendly access to large volumes of snowpack simulations in support of answering critical avalanche forecasting questions like *Which weak layers exist?*, and *How distributed and sensitive to triggering are they?*.

To illustrate the value of our summary perspective on large volumes of snowpack simulations for avalanche research beyond operational avalanche forecasting, Fig. 3 demonstrates how our approach can be used to systematically compare different stability indices that have been used for characterizing instability in simulated profiles. Panels a–e in Fig. 3 visualize the stability distribution of each layer analogously to Panel b in Fig. 2 for the relative threshold sum approach RTA (Monti and Schweizer, 2013), the multi-layered skier stability index SK38ML (Monti et al., 2016), the joint RTA and SK38ML approach (Monti et al., 2014; Morin et al., 2020), the critical crack length (RC) (Richter et al., 2019), and the most recent random forest classifier p_unstable (PU) (Mayer et al., 2022). We classified each stability index into categories, such as *very poor, poor, fair, good*, based on thresholds published in the respective papers. For the two approaches that include SK38ML, we use the most recent thresholds published in Fig. 5 of Morin et al. (2020). Since Richter et al. (2019) derived no thresholds for RC values that correspond to layers with poor stability, we use a threshold for the class *very poor* derived from an unpublished analysis by Mayer et al. (2022) and a threshold for the class *poor* that has been derived from manual observations of critical cracks lengths in unstable layers (Reuter et al., 2015). Not surprisingly, the two related indices TSA (Fig. 2b) and RTA (Fig. 3a) that use purely structural considerations show a very similar pattern. The SK38ML shows a similar pattern to RC, which changes entirely when combined with RTA: potentially unstable weak layers are selected with RTA and then evaluated with SK38ML (Monti et al., 2014; Morin et al., 2020). Since RC is one of the input variables to PU, both are generally similar to each other, while PU substantially reduces the layers with poor stability. Instead of comparing these indices for one simulated profile, our approach allows for valuable large-scale comparisons based on many profiles, which were previously inaccessible.

## 3.2 A representation of the predominant conditions over the course of the season

Since snowpack and avalanche conditions evolve continuously throughout a season, being able to effectively present the evolution of the predominant conditions across forecast regions is critical for supporting forecasters' assessment process and mental models of existing conditions. Our averaging algorithm can be used to represent a temporal perspective of the average conditions in a consistent way by looping it over the course of the season and using the average profile from the previous day as initial condition.

In the time series implementation of our algorithm the height of the snowpack grows over the course of the season by matching the current day individual profiles against the previous day average profile in an open-ended bottom-up alignment approach (for more details, see Herla et al., 2021). This allows new snow layers that are not present in the previous average profile to get stacked on top of the old snow column if more than 50 % of the grid points contain new layers. The amount of

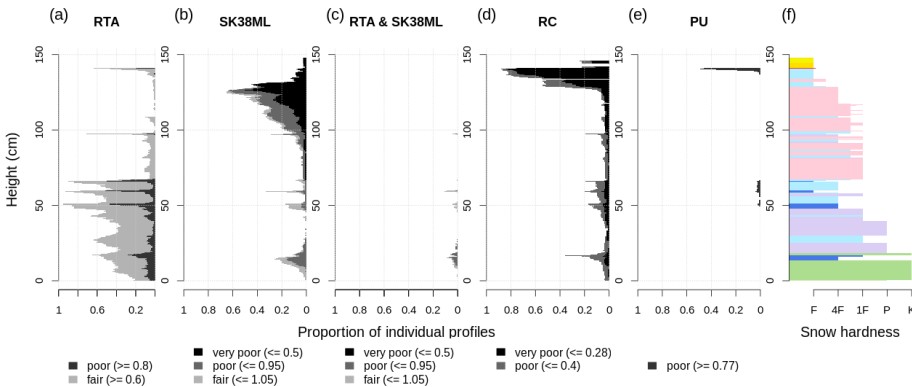

**Figure 3.** The average profile enables users to compare the distributions of different stability indices. Panels (a–e) show the proportion of individual profiles that contain layers with poor stability as diagnosed by the threshold sum approach TSA (i.e., lemons, Schweizer and Jamieson, 2007; Monti et al., 2012, 2014), the relative threshold sum approach RTA (Monti and Schweizer, 2013), the multi-layered skier stability index SK38ML (Monti et al., 2016), the critical crack length (RC) (Richter et al., 2019), and the most recent random forest classifier p_unstable (PU) (Mayer et al., 2022). Panel (f) shows the corresponding average profile (at Jan 20).

new snow in the updated average profile therefore represents the median amounts reported in the profile set. The same effect allows for the growing of thin weak layers at the snow surface.

To capture settlement and melting, the average profile is rescaled after each day if it exceeds the median snow height. Layers that were added on the same day remain unchanged. However, to account for the settlement of freshly buried layers, the upper part of the old snow column is rescaled with a uniform scaling factor. The extent of this rescaled column is adjusted each day based on the median depths of all layers to avoid unrealistic settlements in more deeply buried layers. This scaling routine ensures that the time series of the average profile follows the median snow height and that buried layers align closely with their median depths.

Applying our averaging algorithm in this temporal fashion repeatedly to the same simulated profiles in a forecast region yields a continuous time series of averaged profiles that has a very similar look and feel as the time series of the snowpack evolution at individual grid points but contains information from the entire data set (Fig. 4a). To appreciate the capabilities of the algorithm to capture important summary statistics, study the black lines in Panel a of Fig. 4: the solid line represents the median snow height and follows precisely the height of the average profile; the area between the solid line and the dashed line

represents the median thickness of new snow and is well captured by the corresponding layers of PP and DF shown in yellow; and finally, the dotted lines represent the median depths of several weak layers, which align closely with the red and blue colors highlighting the presence of SH and DH layers in the average profile.

Avalanche forecasters in Canada routinely label weak layers that likely remain hazardous for multiple storm periods with date tags to facilitate effective communication and tracking. Hence, the resulting list of persistent weak layers represents those layers

that the forecasters were most concerned about and that also likely caused avalanches. While a full and detailed validation of our

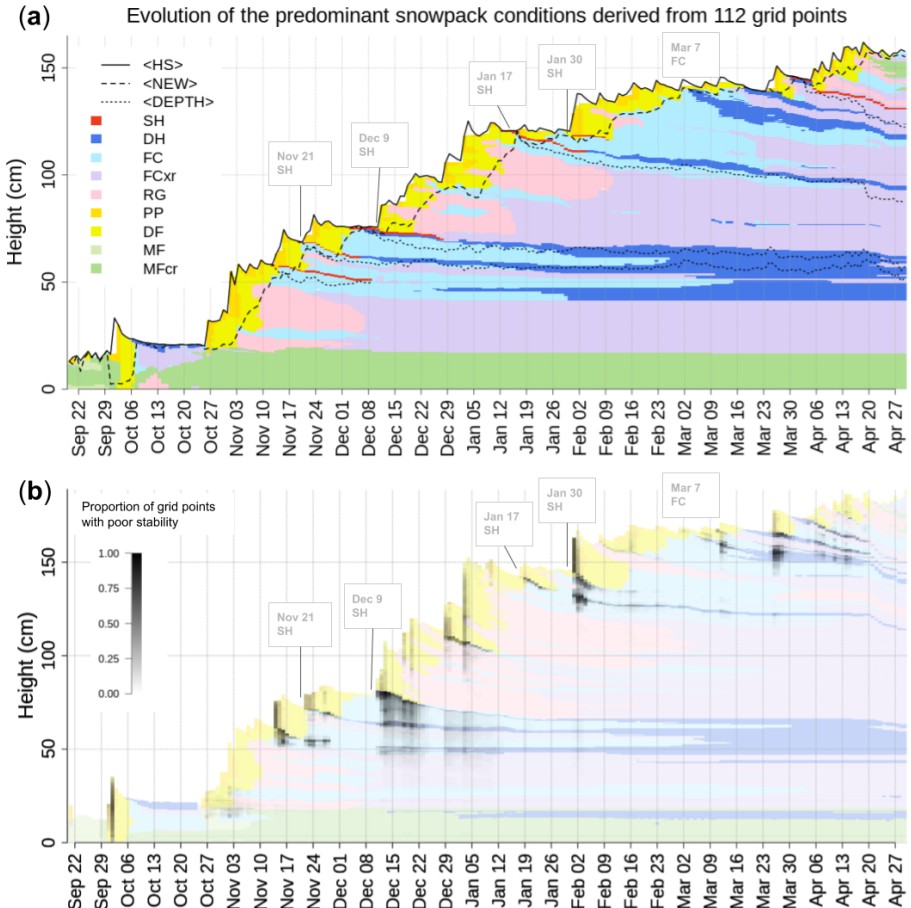

**Figure 4.** (a) Time series of the average snow profile that illustrates the space-averaged evolution of the snow stratigraphy (visualized by snow grain types). The algorithm captures the median total snow height (solid line, <HS>), the median amount of new snow (dashed line, <NEW>), and the median depth of several persistent weak layers (dotted lines, <DEPTH>). (b) The time series of the average profile overplotted with the distribution of grid points that contain layers with poor stability as diagnosed by p_unstable (Mayer et al., 2022); it is therefore the analogon to Fig. 3e in a time series view.

model chain is beyond the scope of this paper (Herla et al., in prep.), the visual comparison of the tracked weak layers and the time series of the average profile presented in Fig. 4 demonstrates that the regionally synthesized snowpack simulations reliably capture the most relevant snowpack features in the region. In an operational context, this visual comparison of simulated and observed weak layer summaries can provide a real-time validation perspective that very efficiently communicates potential discrepancies between modeled weak layers and reality. This allows forecasters to quickly assess when the simulations require cautious interpretation or whether more observations are necessary to verify a yet unobserved weak layer.

In addition to understanding the evolution of the predominant snowpack features, it is equally important for forecasters to understand the evolution of the *stability* of these snowpack features. As discussed earlier, the average profile stores information

about underlying distributions in the profile set, which allows us to visualize the proportion of grid points with poor stability for each layer in the time series of the average profile (Fig. 4b). This visualization takes the concept from Panel e of Fig. 3 to a temporal context and makes it effortless for users to understand temporal trends in the layerwise stability predictions of *all* profiles within the entire data set within a single, very familiar visualization.

## 3.3 Performance of the algorithm during melt season conditions in spring

Physically-based snowpack models are also useful for assessing wet snow avalanche conditions by predicting the depth and the timing of layers accumulating liquid water in the snowpack (Wever et al., 2018). Wever et al. (2018) demonstrate that physically-based snowpack models are capable of simulating the timing of the so-called wetting front within an accuracy of $\pm 1$ day. They also show that the modeled depth of their wetting front correlates well with observed avalanche sizes. While their approach appears promising, its operational application is currently limited to few model grid points because of the lack of spatiotemporal presentation methods that can display this type of complex information effectively. As a consequence, existing operational products for wet snow avalanches are currently limited to bulk indices that represent conditions averaged over the entire snow column (Mitterer et al., 2013; Bellaire et al., 2017; Morin et al., 2020). Hence, wet avalanche forecasting could benefit substantially from data synthesis methods that allow efficient monitoring of the wetting front within regional scale data sets of simulated snow profiles.

To demonstrate the capabilities of our averaging algorithm in supporting wetting and melting conditions, we extracted a set of 46 lower elevation grid points from our data set of simulated snow profiles. While that data set is suited to highlight how our approach can add value to wet avalanche forecasting, operational simulations must consider slope and aspect processes due to their considerable impact on the melting itself. The snowpack at all of these grid points became isothermal before the end of April (Fig. 5d–f show the individual grid points at March 23, March 25, and April 20, respectively). Similarly to the performance of the algorithm with mid-season profiles, the average time series precisely follows the median snow height during melting of the snowpack (Fig. 5c). Furthermore, the averaged profile allows for the monitoring of the median depth of the wetting front as it penetrates into the snowpack (Fig. 5c). In our example, all grid points were entirely sub-freezing and dry before March 23, when warmer air masses (Fig. 5a), cloud cover, and small amounts of liquid precipitation (Fig. 5b) led to the first wetting of the snow surface (Fig. 5c, e). In the consecutive month, the median depth of the wetting front remained constant at roughly 30 cm. A slightly more pronounced rain event on April 19 led to most grid points becoming entirely isothermal (with a frozen surface crust) (Fig. 5c, f). In addition to providing information on the location of the wetting front, distributions or summary statistics of the liquid water content could easily be computed for each averaged layer similarly to extracting or visualizing distributions or summary statistics of the stability of each averaged layer (not shown).

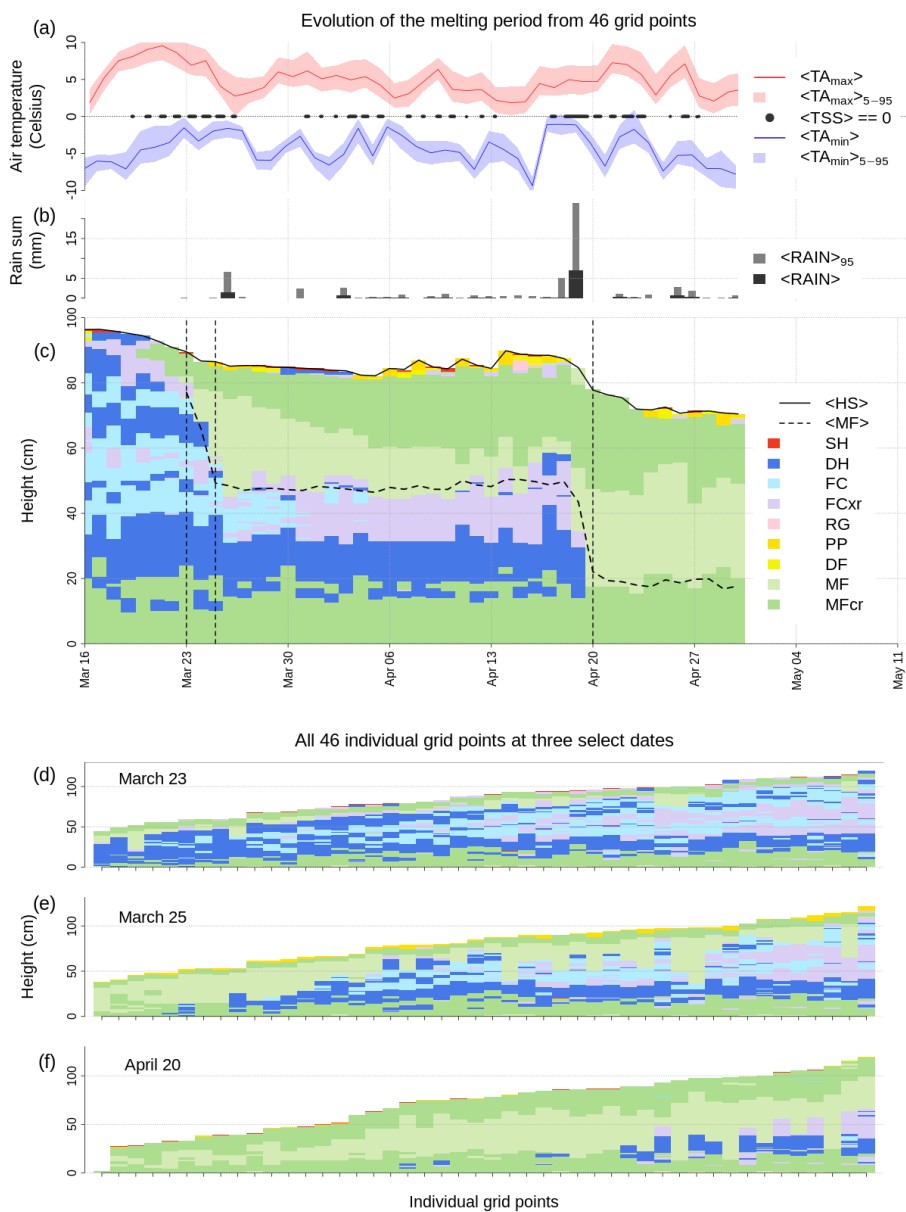

**Figure 5.** Applying the averaging algorithm to wet snow conditions in spring. (a) Daily maximum and minimum air temperatures (median $<...>$, and 5–95 percentile envelopes $<...>_{5-95}$), as well as times when the median snow surface temperature reached zero degrees ($<TSS> == 0$). (b) Rain sums (median and 95 percentile). (c) Time series of the average profile with the median snow height $<HS>$ and the median depth of MF grains, i.e. wetting front, ($<MF>$); the dashed lines represent days for which all individual grid points are shown in panels d–f.

## 4 Performance of the algorithm

### 4.1 Comparison against medoid approach

To quantitatively estimate the performance of the averaging algorithm given the presented data set, we compared the aggregated snow profiles from three different approaches by their root mean squared errors (RMSE):

1. the medoid approach, which identifies the one profile from the profile set that is most similar to all other profiles (Herla et al., 2021),

2. the default averaging approach described in Sect. 2 and 3.1,

3. the timeseries averaging approach described in Sect. 3.2,

We performed this quantitative comparison of methods for every 7th day of the season. The RMSE were computed analogously as described in Sect. 2.

Since the medoid approach follows a simple and transparent concept that has been shown to perform as well or better than more sophisticated sequence aggregation methods (Paparrizos and Gravano, 2015), it represents a meaningful benchmark.
However, the medoid calculations for the 32 days took 28 hours, while the averaging calculations took less than 30 minutes. Despite this immense difference in computational cost, both averaging approaches yielded similar RMSE compared to the medoid approach (Fig. 6). This result suggests that the performance of the aggregation is more influenced by the specifics of the profile set than the peculiarities of the aggregation algorithm. The averaging algorithm presented in this paper therefore performs at least equally well at a much lower computational cost and comes with considerable additional benefits, such as the
capabilities of retrieving underlying distributions and producing consistent time series.

### 4.2 Impacts of the data set and the initial condition on the resulting average profile

The initial condition profile can have substantial influence on the resulting average profile. It is not uncommon that a weak layer that exists in the majority of profiles is not captured in the final average profile if it is not already included in the initial condition profile. It is therefore crucial to select the initial condition profile with care, and to re-run the algorithm for several
different initial conditions as detailed in Sect. 2.

If a prevalent weak layer is not included in the initial condition profile, the odds that the layer will be present in the final average profile depend on the following factors:

– the *prevalence* of the layer in the profile set: the more profiles contain the layer, the more likely it will be included in the final result, because more opportunities exist that the layer is aligned onto the same reference layer.

– the *thickness* of the layer: the thicker the layer, the more likely it will be present in the final result, because it increases the chances of the layer to be aligned. However, this factor is often not relevant, because most weak layers are thin.

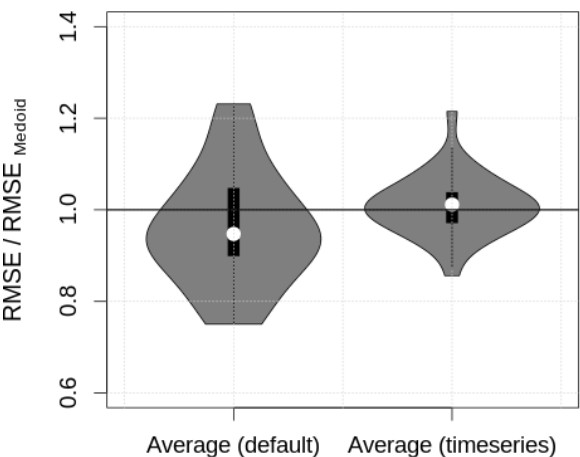

**Figure 6.** Distributions of the root mean squared error (RMSE) of the snow profile averaging methods relative to the RMSE of the Medoid approach for 32 days of the season. Values smaller than 1 suggest that the averaging approaches performed better than the Medoid approach, and vice versa.

    – the *distinctness* of adjacent layers in the profiles: the more distinct or specific the adjacent layers of the weak layer, the more likely it is that it will be in the final result. This is caused by the underlying snow profile alignment algorithm (Herla et al., 2021) that focuses on matching entire layer sequences and not only individual layers. Distinct layer sequences adjacent to weak layers can therefore be thought of as anchor points during layer matching that tremendously increase the odds that an entire group of layers is matched correctly and thus included in the average profile.

While the prevalence of a layer and the characteristics of the adjacent layers are attributes of the data set, the initial condition is the only factor that can be tuned. Since our algorithm automatically picks (multiple) suitable initial conditions by default (see Sect. 2), it is very unlikely that only unsuitable starting conditions are chosen accidentally. However, Fig. 7 explicitly illustrates the effect of the initial condition profile to provide more information on the intricacies of our algorithm.

A scarcely distributed surface hoar layer that is included in 40 % of grid points can be found roughly 20 cm below the new snow within a thick sequence of unspecific bulk layers (Fig. 7a—the layer is emphasized in all panels by slightly more salient and black color). The occurrence frequency threshold to include weak layers in the average profile is set to 30 % in this example. Five out of six initial condition profiles that include that layer (Fig. 7b) lead to average profiles that also contain that layer (Fig. 7c), even though the three influencing factors are all adverse: the layer's prevalence is low (it only exists in a few more profiles than the minimum threshold), it is very thin, and the bulk layer sequences around the weak layer are not distinct and can be found in many other locations of the profiles as well. Panels d and e of Fig. 7 further illustrate the importance of

the presence of the layer of interest in the initial condition profile because all of the average profiles that were initialized with profiles that lacked the surface hoar layer (Fig. 7d) did not include the layer as well. If, however, the surface hoar is adjacent to a distinct crust (Fig. 7f), the resulting average profiles do contain both the crust and surface hoar layer (Fig. 7g) even if they are not present in the initial condition profile. This experiment demonstrates that the odds of a specific layer being present in the final result depend on the interplay of the presented factors and that our routine for the strategic selection of initial conditions is a capable way for employing the algorithm to our best benefit.

### 4.3 Limitations

The quantitative experiments that are presented above demonstrate that the algorithm and the implemented rules produce the desired outcome. While we have not examined the performance of the algorithm in operational avalanche forecasting explicitly, extensive testing by the research team during the development and informal explorations by Avalanche Canada forecasters have shown that the presented DBA approach creates representative snow profiles that summarize the most important snowpack features and highlight the existence of prevalent weak layers and slabs in a meaningful way. However, further explorations are required to better understand the full operational value of our algorithm.

There are three limitations of our algorithm that users should be aware of. The largest source of error in averaging snow profiles originates from the layer matching step. As discussed in Herla et al. (2021), applying the matching algorithm to highly diverse data sets in an unsupervised manner will inevitably produce some alignment inconsistencies and errors since a single hyperparameter setting can naturally not be optimal for the full range of observed conditions. While the impact of this issue is generally negligible for the high-level structure of average profiles, it is important to remember that the extracted distributions of layer and profile characteristics do not represent the truth and need to be evaluated in light of this source of uncertainty. Poor layer matches between snow profiles are more likely if their snow depths differ considerably or their layer sequences show very few common patterns. It is therefore imperative that users judge whether it is meaningful to compute an average profile for a specific set of profiles, and we advise to allow for more margin of error the more diverse the snow profiles are.

Even larger data sets of manual profiles typically contain too few profiles from the same day to sample the considerable spatial variability of the snowpack adequately enough to form a precise mental model of the conditions. Whereas experienced forecasters can account for this when updating their prior mental model with recent observations, it is challenging to fully implement this bayesian reasoning of human forecasters numerically. So, even though it is possible to apply our algorithms to manual profiles in theory, averaging such highly variable data sets comes with its particular challenges that we have not fully considered yet.

To produce a realistically looking time series of average profiles, several algorithmic tweaks are necessary, such as using the previous day's average profile as sole initial condition and rescaling parts of the snow column each day. While these tricks come with the benefits of temporal consistency and computational efficiency, they can introduce unrealistic features in special circumstances. One inconvenience, for example, can occur when new snow falls from one day to the next, and surface hoar forms on top of the new snow. Our explorations have shown that in this situation, the surface hoar layer is often only captured in the average time series once it is buried but not during its formation on the snow surface. Furthermore, since the time series is

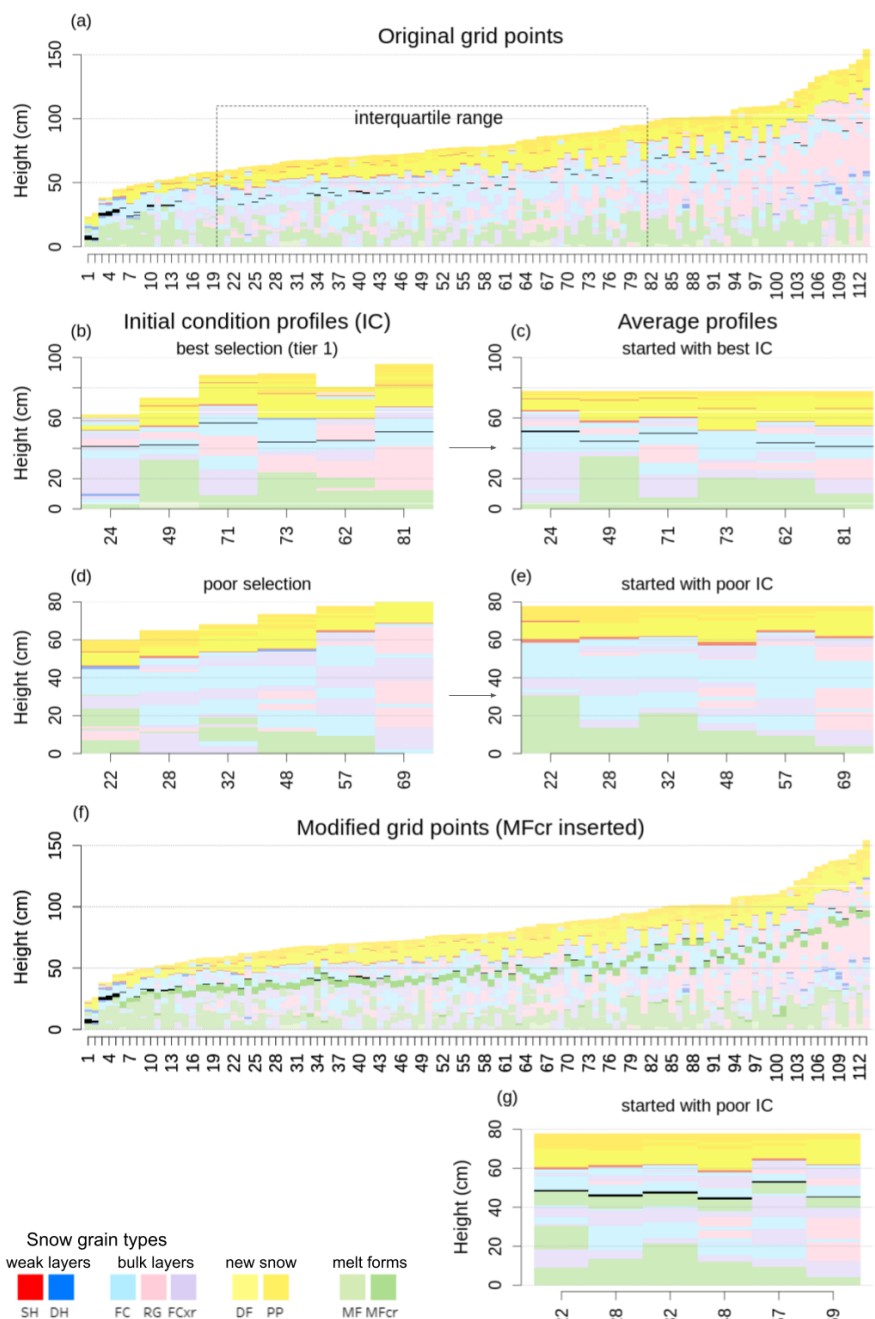

**Figure 7.** An experiment demonstrating the influence of the initial conditions. (a) the grain type sequences of the original grid points with the interquartile range of snow height highlighted. A scarcely distributed weak layer that is located roughly 20 cm below the new snow within the bulk layers is slightly emphasized by more salient and black color in all panels. (b) the grid points chosen by the algorithm as most suitable initial condition profiles (tier 1). (c) the average profiles resulting from the tier 1 starting positions in (b). (d) less suitable starting positions that miss the weak layer within the bulk layers. (e) the average profiles resulting from the suboptimal starting positions in (d), which also miss the particular weak layer. (f) the modified profiles from (a) with an artificially inserted crust in the bulk layers (also emphasized by stronger color). (g) the average profiles resulting from the suboptimal starting positions in (d) when applied to the modified set of profiles in (f); they contain both the crust and the weak layer above. See text for more detailed explanation.

designed to represent several different summary statistics of the profiles (i.e., median snow depth, median layer depths, median new snow amounts), situations can occur when these summary statistics are not completely internally consistent. For example, the surface hoar layer that got buried on April 6 in Fig. 4a happens to primarily exist in profiles that subsequently received above average snowfall amounts (not shown). Consequently, its median layer depth is deeper than what can be represented in the average time series with accurate representations of the median snowfall amounts and the median snow height. In addition to that effect, the lower elevation profiles in the data set became isothermal around April 20, which replaced their original layer sequences with melt forms (not shown). As a result, the median layer depths of the January 17 and April 6 surface hoar layers that were still present in most other profiles increased abruptly, which explains the abrupt drop in the dotted lines in late April. These observations teach us to always examine the results of the presented data exploration tools critically and in the context of the used data set. Other situations might exist that can lead to additional potentially misleading presentations.

## 5  Conclusions

The two snow profile averaging algorithms presented here continue a line of development that aims to make snowpack simulations more accessible and relevant to avalanche warning services and practitioners. Building on the tools introduced by Herla et al. (2021), the presented methods support the analysis of large volumes of snowpack simulations along both space and time by (i) providing quick summary visualizations that help assess the evolution of snow depths, new snow amounts, weak layer and slab combinations, and by (ii) facilitating retrieval of various summary statistics and distributions of layer and profile characteristics.

Without appropriate tools, the operational processing and analysis of simulated stratigraphic information has mainly been restricted to individual grid points, or along either one dimension of space/time. This led to configurations of snowpack simulations in support of avalanche forecasting that are primarily station-based or semi-distributed (Morin et al., 2020). Furthermore, the approaches for evaluating distributed and/or ensemble simulations have so far been limited to bulk properties and summary statistics of the snowpack (Morin et al., 2020; Vernay et al., 2015), which are only of limited interest to avalanche forecasters. By providing summary statistics of layers instead of the entire snow column, our algorithms provide new opportunities for how distributed or ensemble snowpack simulations can be validated and exploited. These new ways of mining available and relevant information aim to inspire new approaches for the operational use of distributed snowpack simulations that are more useful for avalanche forecasting. Furthermore, synthesizing snow profile sets into representative perspectives provides an important and necessary step towards clustering snow stratigraphy information.

While our algorithms open the door for powerful analysis of large data sets of snowpack simulations, there are several limitations that should be considered when applying our methods. It is important to remember that our algorithms are not designed to extract true summaries (i.e., precise average grain size of a particular layer) but rather to facilitate *meaningful explorations* of data sets that are too big for human forecasters to analyze manually.

Even though the impetus for our research was avalanche forecasting, our algorithms might also be of use for other cryospheric researchers interested in the examination of large data sets of snow profiles. Furthermore, the principles behind our DBA approach might also have application for the processing of profiles and time series in other geophysical contexts.

*Code availability.* The presented algorithms are implemented in the R language and environment for statistical computing (R Core Team, 2020) as part of the package `sarp.snowprofile.alignment` (version 1.2.0). The open source package is available from the Comprehensive R Archive Network at https://cran.r-project.org/package=sarp.snowprofile.alignment (Herla et al., 2022a). A static version of the package as well as an annotated demo script to reproduce the figures in this paper are accessible from a permanent repository (Herla et al., 2022b). Our package builds upon the open source packages `dtw` (https://dynamictimewarping.github.io/, last access: 12 January 2022, by Giorgino, 2009), which contains the Dynamic Time Warping implementations, and `sarp.snowprofile` (https://cran.r-project.org/package=sarp.snowprofile, last access: 12 January 2022, by Horton et al., 2020a), which contains basic functionality for reading and manipulating snow profile data.

*Author contributions.* All authors conceptualized the research that FH conducted; FH designed, implemented, and tested the software and wrote the initial manuscript; all authors contributed to reviewing and editing the manuscript; PH supervised the research and acquired the funding.

*Competing interests.* The authors declare that they have no conflict of interest.

*Acknowledgements.* Florian Herla thanks Stephanie Mayer for numerous discussions and for providing her new random forest snow layer stability model. The authors thank Nora Helbig for supervising the review process, as well as the two referees Frank Techel and Christoph Mitterer for their constructive and valuable comments that improved the paper considerably.

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
