# Peer review of "A data exploration tool for averaging and accessing large data sets of snow stratigraphy profiles useful for avalanche forecasting"

_The Cryosphere, 2022_

## Author Comment (AC1)

**Discussion of "Brief communication: A numerical tool for averaging large data sets of snow stratigraphy profiles useful for avalanche forecasting"**

**AUTHOR RESPONSE 1**

**Herla et al.**

**May 10, 2022**

Dear editor, dear referees,

We thank both referees, Frank Techel and Christoph Mitterer, for their supportive reviews, their thoughtful suggestions and their constructive feedback that will help us improve the manuscript. This document provides our responses to their comments and suggestions.

We agree on the importance of the topics that were brought up by the referees and will address them in our revised manuscript. However, since the current manuscript already makes full use of the allowed word count and figure numbers, we kindly ask the editor to change the manuscript type from "Brief Communication" to "Research Article". This will give us the necessary space to adequately address the referees' comments. In the following, we respond to the referee's suggestions in a point-by-point manner, which we will incorporate in our revised manuscript upon the editor's decision on the manuscript type.

**Responses to Referee #1 (Frank Techel)**

**1.1 General Comment**

Referee Comment: *Dear editor, dear authors, the manuscript by Herla et al. introduces a novel method that allows the synthesis of a large number of simulated snow cover simulations resulting in an average profile, which can further be queried if an in-depth analysis is of interest to the user. The proposed method builds upon and expands previous research in this direction. Furthermore, the presented algorithm provides a solution to facilitate the interpretation of snow cover simulations for regional avalanche forecasting.*

*The manuscript is well written, concise, but still easy to follow. The figures are of high quality, supporting the understanding of the described workflow (Fig. 1) and the visualizations obtained with the algorithm (Fig. 2 and 3).*

**Author Comment**: Thank you for your positive and encouraging feedback! We value and appreciate your suggestions and respond to them in a point-by-point manner below.

**1.2 Elaborate on testing of algorithm**

Referee Comment: *On l. 104, the authors state that their testing has shown that the rules applied for initiating the algorithm consistently produced reasonable results. While the described rules (l87-95) do indeed sound plausible, no further detail regarding the testing is provided. - Please elaborate on this testing. For instance, provide a reference if the tests you have made are described elsewhere. What do you mean when you say that "reasonable" average snow profiles are produced? Does reasonable mean that you compared these profiles with observations or is this based on feedback from avalanche forecasters?*

**Author Response:** We tested the algorithm both quantitatively and qualitatively. First, we performed a quantitative comparison of aggregated profiles that were computed with three different approaches:

1. the medoid profile, which identifies the one profile from the profile set that represents the group the best (Herla et al. 2021). While this medoid approach is computationally very expensive and inefficient, it has been shown to perform equally well or better than other more sophisticated and efficient sequence aggregation methods (Paparrizos and Gravano 2015). Besides the algorithm presented in this manuscript, the medoid approach is the only aggregation method that has been applied to snow stratigraphy.

2. the average profile, computed as in Sect. 2 and 3.1 of this manuscript (default algorithm).

3. the average profile, computed as in Sect. 3.2 of this manuscript (timeseries algorithm).

We performed this quantitative comparison of methods for every 7th day of the season. While the averaging calculations took less than 30 minutes, the medoid calculations for the 32 days took 28 hours. Despite this immense difference in computational demand, both averaging approaches yielded very similar root mean squared errors (RMSE) compared to the medoid approach (Fig. 1). This suggests that—in the presented experiment—the performance of the aggregation is more influenced by the specifics of the profile set than the peculiarities of the aggregation algorithm. Given the other disadvantages of the medoid approach (ll. 31–37) and the considerable advantage of the averaging algorithm in making underlying distributions of the profile set accessible (ll. 116–118), the averaging algorithm becomes the obvious winner of the direct comparison.

Besides that quantitative comparison experiment with another profile aggregation method, we qualitatively assessed the value of the algorithm both in an academic context as well in an operational context. Two subsequent and currently ongoing research projects use the algorithm as data exploration tool. First, a large-scale validation study uses the algorithm (timeseries implementation) to provide context to detailed comparisons between human assessment data and snowpack simulations. In that context, the algorithm yielded meaningful results for multiple seasons and different elevation bands (i.e., captured most prevalent weak layers, median quantities like snow height, new snow amounts, etc, and did not yield any obvious erroneous artifacts). While these observations are qualitative for now, that project will generate more quantitative estimates of the capability of the algorithm to capture the predominant weak layers of the underlying profiles. Second, an ongoing clustering project aims to combine clustering methods with the averaging algorithm. In that context the algorithm (default implementation) is used extensively to summarize the conditions of profile clusters and produces aggregations that satisfy our demands for presenting the general snowpack structure on a high level as well as capturing the presence/absence of slab and weak layers. Finally, the algorithm (default implementation) has been used in an operational product at Avalanche Canada during the 2021/22 season to summarize profiles from predefined zones. In that application, the average profile helped forecasters to get a familiar overview of the general snowpack structure in that zone before consulting the

[Figure]

Figure 1: Distributions of the root mean squared error (RMSE) of the three different snow profile aggregation methods for 32 days of the season. In this experiment, the RMSE can range within [0, 1] with better values at the lower end of the scale.

grain type sequences of individual grid points (Simon Horton, personal communication, 2022).

We will elaborate on the testing of the algorithm in our revised manuscript and include the comparison of the averaging algorithm and the medoid approach.

**1.3 Capabilities of the algorithm during wet snow conditions and melting**

Referee Comment: *Section 3.2 and Figure 3 show an example of an average snow profile over the course of a season. This example is helpful as it nicely illustrates the potential of the presented algorithm for the analysis of snow-cover simulations at a regional scale. However, from the perspective of a potential user of this algorithm, it would be useful if you could address the following two points: (1) In this example, the early part of the season is presented, but the melting season is missing. This makes me wonder whether the algorithm works equally well in spring when the snowpack height and the number of simulated layers decrease with increasing wetting. As the first wetting of the snowpack is highly relevant for forecasting wet-snow avalanches, snowpack characteristics like the advance of the wetting front are very important pieces of information (e.g. Wever et al., 2018). - I suggest expanding the average profile shown in Figure 3 well into spring; or, in case the algorithm is less reliable during the melting period, to mention this limitation.*

**Author Response:** Thank you very much for bringing this up!

The algorithm can indeed be used for melt season conditions equivalently as shown for mid-season conditions. Since our operational data archives for the meteorological forcing end in April each year, it is impossible for us to simulate the snowpack until it is fully melted out. However, we extracted a set of grid points from lower elevations to demonstrate the melt season capabilities of the algorithm. All of the newly selected 46 grid points (from below treeline) become isothermal before the end of April (Fig. 2d–f show the individual grid points at March 23, March 25, and April 20, respectively). Similarly to the performance of the algorithm with mid-season

profiles, the average time series precisely follows the median snow height also during melting of the snowpack (Fig. 2c). Furthermore, it also allows for monitoring the wetting of the snowpack, as it continues to follow the median depths of layers (such as the wetting front) (Fig. 2c). In our example, all grid points were entirely sub-freezing before March 23, when warmer air masses (Fig. 2a), cloud cover, and small amounts of liquid precipitation (Fig. 2b) led to the first wetting of the snow surface (Fig. 2c, e). In the consecutive month, the median depth of the wetting front remained constant at roughly 30 cm. A slightly more pronounced rain event on April 19 led to most grid points being entirely isothermal (with a frozen surface crust) (Fig. 2c, f).

We emphasize that our algorithm purely aggregates the information from the underlying grid points. We do not draw any conclusions about the performance of the individual grid points on accurately capturing snow conditions during wetting of the snowpack.

We will demonstrate the algorithm's capabilities of supporting forecasting for wet snow conditions in our revised manuscript.

**1.4 Include summary of observed weak layers**

Referee Comment: *(2) I personally would have greatly appreciated if this example would have been supported with the (observed) weak layer summary in the region. From what I remember, the main author presented such a comparison at a conference last year (Herla et al., 2021), showing that the average profile captures most of the weak layers tracked by the field observers. While not a full validation, this would help the reader to understand that the average snow profile, synthesizing snowpack simulations driven with an NWP model, captures the most important snowpack features in the region.*

**Author Response:** We agree with the referee that a brief summary of the observed weak layers in the region will make the presented example more tangible for readers. We will include it in our revised manuscript by highlighting observed weak layers in Fig. 3 of the original manuscript.

[Figure]

Figure 2: Applying the averaging algorithm to wet snow conditions in spring. (a) Daily maximum and minimum air temperatures (median <...>, and 5–95 percentile envelopes <...>_{5–95}), as well as times when the median snow surface temperature reached zero degrees (<TSS> == 0). (b) Rain sums (median and 95 percentile). (c) Time series of the average profile with the median snow height <HS> and the median depth of MF grains (<MF>); the dashed lines represent times for which all individual grid points are shown in (d–f).

**Responses to Referee #2 (Christoph Mitterer)**

**2.1 General Comment**

Referee Comment: *Dear editor, dear authors, the presented brief communication by Florian Herla and collegues describes the use of a specific averaging technique, the Dynamic Time Warping Barycenter Averaging (DBA) in the field of Dynamic Time Warping (DTW) with pure focus on analysing modelled snow stratigraphy. While the appraoch and methods are not novel, it is the first time that these set of methods was applied to modelled snow stratigraphy and to the field of avalanche forecasting. Large parts of the devleopped DTW method for modelled snow stratigraphy were presented in an earlier manuscript by Herla et al (2021). The focus and added value on the presented manuscript compared to the already published content by Herla et al (2021) is on (1) the newly added averaging technique DBA (section 2), (2) some added features on the layer matching appraoch (lines 73-79) and (3) two newly presented sets of figures (Fig. 2 and 3) for better communicating the obtained results.*

*The text is well written and most of the prestented Figures are clear, easy to understand and enjoyable. Sometimes explanations are a bit to short and due to the nature of a brief communication explanations are sometimes not easy to grasp for an uniformed reader. In addiation, I would suggest improving Figure 1.*

*Even though parts of the content were already described in Herla et al (2021), I like the idea of this brief communication since the authors focus more on the quality of the results while within the other publication the architecture of the algorithm covered most parts of the reading. Nevertheless, I would expect a little more quantitative presentation on some of the descriptions, which leads me to my four general comments that may improve the quality of the manuscript:* [continued below]

**Author Comment**: Thank you for your encouraging review and your thoughts on improving the manuscript, we much appreciate it. Upon changing of the manuscript type, we will be able to elaborate on some explanations in slightly more detail and context. Thank you for bringing this up!

**2.2 Improve Figure 1**

Referee Comment: *As stated, Figure 1 is a bit confusing and hard to understand. Could you maybe use less profiles in between and describe the workflow a bit more in detail within the graph. In addition, add some more description within the caption.*

**Author Response:** Yes, we will remove some profiles to make the figure less overwhelming, and we will describe the workflow within the caption.

**2.3 Elaborate on influence of initial conditions**

Referee Comment: *You state that for the DBA it is essential to start the interation by choosing initial condition profiles strategically (line 89). How influential is that condition of the initial profile? Or with other words, if I miss to chose my starting position carefully, does the algorithm support me and is able to find weak layers that I just missed when picking the starting conditions. Can you quantify that by adding some noise to your initial profile?*

**Author Response:** The initial condition profile can indeed have substantial influence in the resulting average profile. It is well possible that a weak layer that exists in the majority of profiles but not in the initial condition profile is not captured in the final result. It is therefore crucial to select the initial condition profile with care, and to re-run the algorithm for several different

initial conditions (l. 104). If a prevalent weak layer is not included in the initial condition profile, the odds that this layer will be present in the final average profile depend on (with increasing influence)

- the distribution of this layer in the profile set: The more profiles contain this layer, the more likely it is that it will be present in the final result, because more opportunities exist that this layer will be aligned onto the same reference layer.

- the thickness of this layer: The thicker the layer, the more likely it is that it will be present in the final result, because of the same reason as above. Note that this factor is often invariant, because many weak layers are thin.

- the location of this layer in the profiles: The more particular or specific the adjacent layers of the weak layer, the more likely it is that it will be in the final result. The underlying snow profile alignment algorithm (Herla et al. 2021) that the averaging algorithm builds upon optimizes not only the matching of individual layers, but the matching of entire layer sequences. Particular layer sequences adjacent to individual weak layers can therefore be thought of as anchor points during layer matching that tremendously increase the odds that the entire neighborhood of layers will be matched correctly and thus included in the average profile.

So, the main factors that influence whether a thin weak layer is included in the average profile are (i) distribution of the layer, (ii) the specificity of neighboring layer sequences, and (iii) the initial condition profile. While the first two factors are attributes of the data set, the initial condition is the only factor that can be tuned.

For example, assume a weak layer exists in 40 % of grid points within a thick sequence of *unspecific* bulk layers. If the occurrence frequency threshold to include that weak layer in the average profile (ll. 83–84) is set to 30 % and the initial condition profile does *not* contain that layer, then the odds that this layer will be present in the average profile tend to zero, because all four important factors are adverse: the layer only exists in a few more profiles than the minimum threshold, it is very thin, the bulk layer sequences around the weak layer are not specific but can be found in many other locations of the profiles as well, and the initial conditions do not contain the layer. However, either if

- that particular layer *is* present in the initial condition profile, or

- that particular layer is located next to a particular snowpack feature, for example a crust that is present in most grid points,

then the odds that this layer is present in the final result are a lot higher. Figure 3 visualizes the example and illustrates these statements: A scarcely distributed surface hoar layer can be found roughly 20 cm below the new snow within the bulk layers (Fig. 3a—layer emphasized by slightly more salient color). Five out of six initial condition profiles that do contain that layer (Fig. 3b) lead to average profiles that also contain that layer (Fig. 3c). By contrast, using an initial condition profile that misses that layer (Fig. 3d) results in an average profile that also does not contain that layer (Fig. 3e). If, however, the grain type sequence around the surface hoar layer were more particular—we model that by artificially ingesting a crust layer below the surface hoar layer, or where non-existent somewhere within the bulk layers (Fig. 3f)—the resulting average profile does contain both crust and surface hoar layer (Fig. 3g), although both were not present in the initial condition profile (Fig. 3d).

We emphasize that our algorithm per default picks (multiple) suitable initial conditions by itself, so accidentally picking only unsuitable starting conditions is highly unlikely. As stated

in the manuscript, it searches for appropriate profiles within the interquartile range of snow height (Fig. 3a), and selects profiles with an above number of weak layers in as many depth ranges as possible (l. 90). That means, it organizes the profiles into several tiers by number of occupied depth ranges[1] and by total number of weak layers. Tier 1 contains all profiles with the maximum number of occupied depth ranges and the maximum total number of weak layers. Tier 2 contains the remaining profiles with the maximum number of occupied depth ranges and an above-average number of weak layers. Tier 3 contains all remaining profiles with the sub-maximum number of occupied depth ranges and an above average number of weak layers. Depending on how many initial conditions are requested by the user the algorithm picks profiles from the three tiers. While the algorithm can pick the appropriate starting conditions by itself, the user can modify how the algorithm should search for weak layers by labeling layers of interest (ll. 75-79). In other words, structural considerations (such as grain type, grain size, etc) can be combined with a number of popular stability indices given that they are available in the profiles. Currently implemented are threshold sums TSA or RTA (Schweizer and Jamieson 2007; Monti and Schweizer 2013), SK38 (Monti et al. 2016), critical crack length (Richter et al. 2019), and p_unstable (Mayer et al. 2022).

We will include in the revised manuscript

- a more detailed description of how initial conditions are picked by the algorithm and explain users' options for customization,

- a more detailed description of the factors influencing the resulting average profile and emphasize the role of the initial condition.
* * *
[1]The depth ranges are per default set to [0, 30), [30, 80), [80, 150), [150, Inf) (cm), but can be modified by the user.

[Figure]

Figure 3: An experiment demonstrating the influence of the initial conditions. (a) the grain type sequences of the original grid points with the interquartile range (IQR) of snow height highlighted; a scarcely distributed weak layer that is located roughly 20 cm below the new snow within the bulk layers is slightly emphasized by more salient color. (b) the grid points chosen by the algorithm as most appropriate initial condition profiles (IC, tier 1). (c) the average profiles (AVG) resulting from the tier 1 starting positions in (b). (d) a less suitable starting position that misses the weak layer within the bulk layers. (e) the average profile resulting from the suboptimal starting position in (d), which also misses the particular weak layer. (f) the modified profiles from (a) with an artificially inserted crust in the bulk layers (again emphasized by color). (g) the average profile resulting from the suboptimal starting position in (d) when applied to the modified set of profiles in (f); it does contain both the crust and the weak layer above. See text for more detailed explanation.

**2.4 Elaborate on testing of algorithm**

Referee Comment: *Related to that are your statements on the testing. I would like to see some more quantitative results and more in-depth description on how you did that. Reading phrases like (…)consistently produce reasonable average snow profiles suitable for avalanche forecasting. (Line 105), are with low support and not helpful for the interested reader or an avalanche forecaster that wants to apply your findings. In addition, I would be curious what you think is suitable for avalanche forecasting and what is not ;-).*

**Author Response:** This has been answered in detail in Author Comment 1.2.

**2.5 Include better stability index**

Referee Comment: *I like Fig. 2 very much. It will be very helpful in daily routines of avalanche forecasting centers. However, I have some issues with how the content of Fig. 2b was produced. You basically applied the approach by Schweizer and Jamieson (2007) which turned out to be inpropriate or at least less helpful when applied to simulated snow cover data (Monti and Schweizer, 2013). Main reason for that is the fact that the thresholds by Schweizer and Jamieson (2007) were obtained with statistics based on observed snow stratigraphy parameters which may differ compared to simulated ones (especially grain size). That's why Monti and Schweizer (2013) introduced the relative threshold sum appraoch and I would love to see if there are particular differences for the presented example. In fact, I would expect, e.g. the facets below the thick layer of RGs (I assume this to be the slab) to give more indication towards instability. This in turn would give you the option to included FCs as weak layers as well. At the moment the representation of Fig. 2b is heavily driven by grain size only, since the used underlaying snow cover model classifies the weak layer DH and SH mainly based on their size.*

**Author Response:** It is very encouraging to hear that you think our approach of making underlying distributions in the data set accessible will be helpful for operational avalanche forecasting. With regards to the threshold sum approach (TSA) by Schweizer and Jamieson (2007), we agree with you in that it is not a state-of-the-art stability index for simulated profiles. It is, however, a conceptually straightforward approach that is very tangible to many practitioners due to its application in the field. Since our figure aims at presenting the general capabilities of our algorithm, we believe it is most valuable to keep the complexity of the presented stability assessment low at this point in the paper. To address your suggestion of comparing the presented TSA approach to more sophisticated stability indices, we will follow up our general overview figure with Fig. 4. This new figure will highlight very strongly that our approach of making underlying distributions of the data set accessible is not confined to a single and particular property, but can be applied to any available variable in the user's data set. Furthermore, it will visualize that our averaging approach creates an opportunity for systematically comparing stability indices on a large-scale (i.e., for *many* profiles instead of few select ones) and for designing sensitivity studies that assess impacts on the layer level.

Since the recent development by Mayer et al. (2022) tremendously improves upon the temporal resolution of previously existing stability indices for simulated profiles, we will also visualize the proportion of grid points that promote poor layer stability in the time series of the average profile (Fig. 5). This visualization takes the concept from Fig. 4f to a temporal context and makes it effortless for users to understand temporal trends in the layerwise stability predictions of *many* profiles.

[Figure]

Figure 4: The average profile enables users to compare the distributions of different stability indices. Panels (a–e) show the proportion of individual profiles that contain layers with poor stability as diagnosed by the threshold sum approach TSA (i.e., lemons, Schweizer and Jamieson 2007; Monti et al. 2012, 2014), the relative threshold sum approach RTA (Monti and Schweizer 2013), the multi-layered skier stability index SK38ML (Monti et al. 2016), the critical crack length (RC) (Richter et al. 2019), and the most recent random forest classifier p_unstable (PU) (Mayer et al. 2022). Panel (f) shows the according average profile (at Jan 20).

[Figure]

Figure 5: The time series of the average profile as in Fig. 3 of the manuscript, but overplotted with the distribution of grid points that contain layers with poor stability as diagnosed by p_unstable (Mayer et al. 2022); it is therefore the analogon to Fig. 4e in a time series view.

**2.6 Details of snowpack model**

Referee Comment: *Can you please give some more insights of the model behind the modelled snow stratigraphy data? Are you using SNOWPACK or Crocus?*

**Author Response:** We use a weather and snowpack model chain. Our weather model HRDPS (Milbrandt et al. 2016) has a 2.5 km resolution and provides the meteorological forcing for the model SNOWPACK (Bartelt et al. 2002; Lehning et al. 2002b,a).

While this fact may be interesting for the reader, it has no relevance for the averaging tool we present. In other words, our tool can be applied to any (simulated) snow profile irrespective of its source model. We will make sure to include this information in our revision. In fact, our tool can also be applied to sets of manual profiles. We have not yet had an appropriate data set for that application and have not yet attempted it, though.

**2.7 Clarify capabilities in wet snow conditions**

Referee Comment: *The algorithm seems to work dry snow conditions only? Can you comment on that?*

**Author Response:** No, it works equally well for wet snow and melting conditions in spring, see our comment in 1.3 and Fig. 2.

**References**

Bartelt, P., Lehning, M., Bartelt, P., Brown, B., Fierz, C., and Satyawali, P.: A physical SNOWPACK model for the Swiss avalanche warning: Part I: Numerical model, Cold Regions Science and Technology, 35, 123–145, https://doi.org/10.1016/s0165-232x(02)00074-5, 2002.

Herla, F., Horton, S., Mair, P., and Haegeli, P.: Snow profile alignment and similarity assessment for aggregating, clustering, and evaluating of snowpack model output for avalanche forecasting, Geoscientific Model Development, 14, 239–258, https://doi.org/10.5194/gmd-14-239-2021, 2021.

Lehning, M., Bartelt, P., Brown, B., and Fierz, C.: A physical SNOWPACK model for the Swiss avalanche warning Part III: Meteorological forcing, thin layer formation and evaluation, Cold Regions Science and Technology, 35, 169–184, https://doi.org/10.1016/S0165-232X(02)00072-1, 2002a.

Lehning, M., Bartelt, P., Brown, B., Fierz, C., and Satyawali, P.: A physical SNOWPACK model for the Swiss avalanche warning Part II. Snow microstructure, Cold Regions Science and Technology, 35, 147–167, https://doi.org/10.1016/S0165-232X(02)00073-3, 2002b.

Mayer, S., Herwijnen, A. V., Techel, F., and Schweizer, J.: A random forest model to assess snow instability from simulated snow stratigraphy, pp. 1–39, https://doi.org/10.5194/tc-2022-34, 2022.

Milbrandt, J. A., Bélair, S., Faucher, M., Vallée, M., Carrera, M. L., and Glazer, A.: The pan-canadian high resolution (2.5 km) deterministic prediction system, Weather and Forecasting, 31, 1791–1816, https://doi.org/10.1175/WAF-D-16-0035.1, 2016.

Monti, F. and Schweizer, J.: A relative difference approach to detect potential weak layers within a snow profile, in: Proceedings of the 2013 International Snow Science Workshop, Grenoble, France, pp. 339–343, URL `https://arc.lib.montana.edu/snow-science/item.php?id=1861`, 2013.

Monti, F., Cagnati, A., Valt, M., and Schweizer, J.: A new method for visualizing snow stability profiles, Cold Regions Science and Technology, 78, 64–72, https://doi.org/10.1016/j.coldregions.2012.02.005, 2012.

Monti, F., Schweizer, J., and Fierz, C.: Hardness estimation and weak layer detection in simulated snow stratigraphy, Cold Regions Science and Technology, 103, 82–90, https://doi.org/10.1016/j.coldregions.2014.03.009, 2014.

Monti, F., Gaume, J., van Herwijnen, A., and Schweizer, J.: Snow instability evaluation: calculating the skier-induced stress in a multi-layered snowpack, Natural Hazards and Earth System Sciences, 16, 775–788, https://doi.org/10.5194/nhess-16-775-2016, 2016.

Paparrizos, J. and Gravano, L.: k-shape: Efficient and accurate clustering of time series, in: Proceedings of the 2015 ACM SIGMOD International Conference on Management of Data, pp. 1855–1870, ACM, 2015.

Richter, B., Schweizer, J., Rotach, M. W., and Van Herwijnen, A.: Validating modeled critical crack length for crack propagation in the snow cover model SNOWPACK, The Cryosphere, 13, 3353–3366, https://doi.org/10.5194/tc-13-3353-2019, 2019.

Schweizer, J. and Jamieson, J. B.: A threshold sum approach to stability evaluation of manual snow profiles, Cold Regions Science and Technology, 47, 50–59, https://doi.org/10.1016/j.coldregions.2006.08.011, 2007.

---

## Author Response (AR1)

Discussion of "A data exploration tool for averaging and accessing large
data sets of snow stratigraphy profiles useful for avalanche forecasting"

**Author response 2**

Herla et al.

June 25, 2022

Dear editor, dear referees,

We thank both referees, Frank Techel and Christoph Mitterer, for their supportive reviews, their thoughtful suggestions and their constructive feedback that helped us improve the manuscript considerably. This document provides the changes we have made to the manuscript prompted by the reviewers' comments and suggestions.

We thank the editor for changing our manuscript type from "Brief Communication" to "Research Article". This gave us the necessary space to adequately address the referees' comments.

In the following, we highlight in a point-by-point manner how we addressed the referee's suggestions. Given page and line numbers refer to the track-changes document of our revised manuscript. In addition to the changes emphasized in this Author Response, we also edited the entire manuscript in detail to improve clarity and readability.

**Responses to Referee #1 (Frank Techel)**

**1.1 General Comment**

Referee Comment: *Dear editor, dear authors, the manuscript by Herla et al. introduces a novel method that allows the synthesis of a large number of simulated snow cover simulations resulting in an average profile, which can further be queried if an in-depth analysis is of interest to the user. The proposed method builds upon and expands previous research in this direction. Furthermore, the presented algorithm provides a solution to facilitate the interpretation of snow cover simulations for regional avalanche forecasting.*

*The manuscript is well written, concise, but still easy to follow. The figures are of high quality, supporting the understanding of the described workflow (Fig. 1) and the visualizations obtained with the algorithm (Fig. 2 and 3).*

**Author Comment**: Thank you for your positive and encouraging feedback! We value and appreciate your suggestions and respond to them in a point-by-point manner below.

**1.2 Elaborate on testing of algorithm**

Referee Comment: *On l. 104, the authors state that their testing has shown that the rules applied for initiating the algorithm consistently produced reasonable results. While the described rules (l87-95) do indeed sound plausible, no further detail regarding the testing is provided. - Please elaborate on this testing. For instance, provide a reference if the tests you have made are described elsewhere. What do you mean when you say that "reasonable" average snow profiles are produced? Does reasonable mean that you compared these profiles with observations or is this based on feedback from avalanche forecasters?*

**Author Response:** In the revised manuscript, we elaborate on the testing of the algorithm by adding the new Section 4.1, *Comparison against medoid approach* (P13 L270):

> "To quantitatively estimate the performance of the averaging algorithm given the presented data set, we compared the aggregated snow profiles from three different approaches by their root mean squared errors (RMSE):
>
> 1. the medoid approach, which identifies the one profile from the profile set that is most similar to all other profiles (Herla et al. 2021),
>
> 2. the default averaging approach described in Sect. 2 and 3.1,
>
> 3. the timeseries averaging approach described in Sect. 3.2,
>
> We performed this quantitative comparison of methods for every 7th day of the season. The RMSE were computed analogously as described in Sect. 2.
>
> Since the medoid approach follows a simple and transparent concept that has been shown to perform as well or better than more sophisticated sequence aggregation methods (Paparrizos and Gravano 2015), it represents a meaningful benchmark. However, the medoid calculations for the 32 days took 28 hours, while the averaging calculations took less than 30 minutes. Despite this immense difference in computational cost, both averaging approaches yielded similar RMSE compared to the medoid approach (Fig. 6). This result suggests that the performance of the aggregation is more influenced by the specifics of the profile set than the peculiarities of the aggregation algorithm. The averaging algorithm presented in this paper therefore performs at least equally well at a much lower computational cost and comes with considerable additional benefits, such as the capabilities of retrieving underlying distributions and producing consistent time series."

Furthermore, we added the following new paragraph (P15 L322):

> "While we have not examined the performance of the algorithm in operational avalanche forecasting explicitly, extensive testing by the research team during the development and informal explorations by Avalanche Canada forecasters have shown that the presented DBA approach creates representative snow profiles that summarize the most important snowpack features and highlight the existence of prevalent weak layers and slabs in a meaningful way. However, further explorations are required to better understand the full operational value of our algorithm."

**1.3 Capabilities of the algorithm during wet snow conditions and melting**

Referee Comment: *Section 3.2 and Figure 3 show an example of an average snow profile over the course of a season. This example is helpful as it nicely illustrates the potential of the presented algorithm for the analysis of snow-cover simulations at a regional scale. However, from the perspective of a potential user of this algorithm, it would be useful if you could address the following*

*two points: (1) In this example, the early part of the season is presented, but the melting season is missing. This makes me wonder whether the algorithm works equally well in spring when the snowpack height and the number of simulated layers decrease with increasing wetting. As the first wetting of the snowpack is highly relevant for forecasting wet-snow avalanches, snowpack characteristics like the advance of the wetting front are very important pieces of information (e.g. Wever et al., 2018). - I suggest expanding the average profile shown in Figure 3 well into spring; or, in case the algorithm is less reliable during the melting period, to mention this limitation.*

**Author Response:** Thank you very much for bringing this up! We added the new Section 3.3, *Performance of the algorithm during melt season conditions in spring* (P11 L245):

"Physically-based snowpack models are also useful for assessing wet snow avalanche conditions by predicting the depth and the timing of layers accumulating liquid water in the snowpack (Wever et al. 2018). Wever et al. (2018) demonstrate that physically-based snowpack models are capable of simulating the timing of the so-called wetting front within an accuracy of $\pm 1$ day. They also show that the modeled depth of their wetting front correlates well with observed avalanche sizes. While their approach appears promising, its operational application is currently limited to few model grid points because of the lack of spatiotemporal presentation methods that can display this type of complex information effectively. As a consequence, existing operational products for wet snow avalanches are currently limited to bulk indices that represent conditions averaged over the entire snow column (Mitterer et al. 2013; Bellaire et al. 2017; Morin et al. 2020). Hence, wet avalanche forecasting could benefit substantially from data synthesis methods that allow efficient monitoring of the wetting front within regional scale data sets of simulated snow profiles.

To demonstrate the capabilities of our averaging algorithm in supporting wetting and melting conditions, we extracted a set of 46 lower elevation grid points from our data set of simulated snow profiles. The snowpack at all of these grid points became isothermal before the end of April (Fig. 5d–f show the individual grid points at March 23, March 25, and April 20, respectively). Similarly to the performance of the algorithm with mid-season profiles, the average time series precisely follows the median snow height during melting of the snowpack (Fig. 5c). Furthermore, the averaged profile allows for the monitoring of the median depth of the wetting front as it penetrates into the snowpack (Fig. 5c). In our example, all grid points were entirely sub-freezing and dry before March 23, when warmer air masses (Fig. 5a), cloud cover, and small amounts of liquid precipitation (Fig. 5b) led to the first wetting of the snow surface (Fig. 5c, e). In the consecutive month, the median depth of the wetting front remained constant at roughly 30 cm. A slightly more pronounced rain event on April 19 led to most grid points becoming entirely isothermal (with a frozen surface crust) (Fig. 5c, f). In addition to providing information on the location of the wetting front, distributions or summary statistics of the liquid water content could easily be computed for each averaged layer similarly to extracting or visualizing distributions or summary statistics of the stability of each averaged layer (not shown).

Similarly to extracting or visualizing distributions or summary statistics of the stability of each averaged layer, distributions or summary statistics of the liquid water content could easily be computed for each averaged layer (not shown)."

**1.4 Include summary of observed weak layers**

Referee Comment: *(2) I personally would have greatly appreciated if this example would have been supported with the (observed) weak layer summary in the region. From what I remember,*

*the main author presented such a comparison at a conference last year (Herla et al., 2021), showing that the average profile captures most of the weak layers tracked by the field observers. While not a full validation, this would help the reader to understand that the average snow profile, synthesizing snowpack simulations driven with an NWP model, captures the most important snowpack features in the region.*

**Author Response:** We agree with the referee and added observed weak layers in Fig. 4 of the revised manuscript. We also added the following new paragraph (P9 L229):

> "Avalanche forecasters in Canada routinely label weak layers that likely remain hazardous for multiple storm periods with date tags to facilitate effective communication and tracking. Hence, the resulting list of persistent weak layers represents those layers that the forecasters were most concerned about and that also likely caused avalanches. While a full and detailed validation of our model chain is beyond the scope of this paper (Herla et al., in prep.), the visual comparison of the tracked weak layers and the time series of the average profile presented in Fig. 4 demonstrates that the regionally synthesized snowpack simulations reliably captures the most relevant snowpack features in the region. In an operational context, this visual comparison of simulated and observed weak layer summaries can provide a real-time validation perspective that very efficiently communicates potential discrepancies between modeled weak layers and reality. This allows forecasters to quickly assess when the simulations require cautious interpretation or whether more observations are necessary to verify a yet unobserved weak layer."

**Responses to Referee #2 (Christoph Mitterer)**

**2.1 General Comment**

Referee Comment: *Dear editor, dear authors, the presented brief communication by Florian Herla and collegues describes the use of a specific averaging technique, the Dynamic Time Warping Barycenter Averaging (DBA) in the field of Dynamic Time Warping (DTW) with pure focus on analysing modelled snow stratigraphy. While the appraoch and methods are not novel, it is the first time that these set of methods was applied to modelled snow stratigraphy and to the field of avalanche forecasting. Large parts of the devleopped DTW method for modelled snow stratigraphy were presented in an earlier manuscript by Herla et al (2021). The focus and added value on the presented manuscript compared to the already published content by Herla et al (2021) is on (1) the newly added averaging technique DBA (section 2), (2) some added features on the layer matching appraoch (lines 73-79) and (3) two newly presented sets of figures (Fig. 2 and 3) for better communicating the obtained results.*

*The text is well written and most of the prestented Figures are clear, easy to understand and enjoyable. Sometimes explanations are a bit to short and due to the nature of a brief communication explanations are sometimes not easy to grasp for an uniformed reader. In addiation, I would suggest improving Figure 1.*

*Even though parts of the content were already described in Herla et al (2021), I like the idea of this brief communication since the authors focus more on the quality of the results while within the other publication the architecture of the algorithm covered most parts of the reading. Nevertheless, I would expect a little more quantitative presentation on some of the descriptions, which leads me to my four general comments that may improve the quality of the manuscript:* [continued below]

**Author Comment**: Thank you for your encouraging review and your thoughts on improving the manuscript, we much appreciate it. Since changing of the manuscript type allowed us to elaborate more, we added more detail and context at several locations throughout the manuscript. Thank you for bringing this up!

**2.2 Improve Figure 1**

Referee Comment: *As stated, Figure 1 is a bit confusing and hard to understand. Could you maybe use less profiles in between and describe the workflow a bit more in detail within the graph. In addition, add some more description within the caption.*

**Author Response:** Yes, we removed some profiles to make Fig. 1 less overwhelming, and we changed the caption of the figure to include the workflow of the algorithm.

**2.3 Elaborate on influence of initial conditions**

Referee Comment: *You state that for the DBA it is essential to start the interation by choosing initial condition profiles strategically (line 89). How influential is that condition of the initial profile? Or with other words, if I miss to chose my starting position carefully, does the algorithm support me and is able to find weak layers that I just missed when picking the starting conditions. Can you quantify that by adding some noise to your initial profile?*

**Author Response:** We added more details on how our algorithm automatically picks strategically meaningful initial conditions (P3 L109):

[revised manuscript text omitted]

**2.4 Elaborate on testing of algorithm**

Referee Comment: *Related to that are your statements on the testing. I would like to see some more quantitative results and more in-depth description on how you did that. Reading phrases like (...)consistently produce reasonable average snow profiles suitable for avalanche forecasting. (Line 105), are with low support and not helpful for the interested reader or an avalanche forecaster that wants to apply your findings. In addition, I would be curious what you think is suitable for avalanche forecasting and what is not ;-).*

**Author Response:** This has been answered in detail in Author Comment 1.2.

**2.5 Include better stability index**

Referee Comment: *I like Fig. 2 very much. It will be very helpful in daily routines of avalanche forecasting centers. However, I have some issues with how the content of Fig. 2b was produced. You basically applied the approach by Schweizer and Jamieson (2007) which turned out to be inpropriate or at least less helpful when applied to simulated snow cover data (Monti and Schweizer, 2013). Main reason for that is the fact that the thresholds by Schweizer and Jamieson (2007) were obtained with statistics based on observed snow stratigraphy parameters which may differ compared to simulated ones (especially grain size). That's why Monti and Schweizer (2013) introduced the relative threshold sum appraoch and I would love to see if there are particular differences for the presented example. In fact, I would expect, e.g. the facets below the thick layer of RGs (I assume this to be the slab) to give more indication towards instability. This in turn would give you the option to included FCs as weak layers as well. At the moment the representation of Fig. 2b is heavily driven by grain size only, since the used underlaying snow cover model classifies the weak layer DH and SH mainly based on their size.*

**Author Response:** It is very encouraging to hear that you think our approach of making underlying distributions in the data set accessible will be helpful for operational avalanche forecasting. With regards to the threshold sum approach (TSA) by Schweizer and Jamieson (2007), we agree with you in that it is not a state-of-the-art stability index for simulated profiles. It is, however, a conceptually straightforward approach that is very tangible to many practitioners due to its application in the field. Since our figure aims at presenting the general capabilities of our algorithm, we believe it is most valuable to keep the complexity of the presented stability assessment low at this point in the paper. To address your suggestion of comparing the presented TSA approach to more sophisticated stability indices, we follow up our general overview figure with Fig. 3. This new figure highlights very strongly that our approach of making underlying distributions of the data set accessible is not confined to a single and particular property, but can be applied to any available variable in the user's data set. We also added the following new

paragraph to describe the new figure (P6 L183):

> "To illustrate the value of our summary perspective on large volumes of snowpack simulations for avalanche research beyond operational avalanche forecasting, Fig. 3 demonstrates how our approach can be used to systematically compare different stability indices that have been used for characterizing instability in simulated profiles. Panels a–e in Fig. 3 visualize the stability distribution of each layer analogously to Panel b in Fig. 2 for the relative threshold sum approach RTA (Monti and Schweizer 2013), the multi-layered skier stability index SK38ML (Monti et al. 2016), the joint RTA and SK38ML approach (Monti et al. 2014; Morin et al. 2020), the critical crack length (RC) (Richter et al. 2019), and the most recent random forest classifier p_unstable (PU) (Mayer et al. 2022). We classified each stability index into categories, such as *very poor, poor, fair, good*, based on thresholds published in the respective papers. For the two approaches that include SK38ML, we use the most recent thresholds published in Fig. 5 of Morin et al. (2020). Since Richter et al. (2019) derived no thresholds for RC values that correspond to layers with poor stability, we use a threshold for the class *very poor* derived from an unpublished analysis by Mayer et al. (2022) and a threshold for the class *poor* that has been derived from manual observations of critical cracks lengths in unstable layers (Reuter et al. 2015). Not surprisingly, the two related indices TSA (Fig. 2b) and RTA (Fig. 3a) that use purely structural considerations show a very similar pattern. The SK38ML shows a similar pattern to RC, which changes entirely when combined with RTA: potentially unstable weak layers are selected with RTA and then evaluated with SK38ML (Monti et al. 2014; Morin et al. 2020). Since RC is one of the input variables to PU, both are generally similar to each other, while PU substantially reduces the layers with poor stability. Instead of comparing these indices for one simulated profile, our approach allows for valuable large-scale comparisons based on many profiles, which were previously inaccessible."

Furthermore, we added Panel b to Fig. 4, which visualizes the proportion of grid points that promote poor layer stability in the time series of the average profile, and added the following new paragraph (P9 L239):

> "In addition to understanding the evolution of the predominant snowpack features, it is equally important for forecasters to understand the evolution of the *stability* of these snowpack features. As discussed earlier, the average profile stores information about underlying distributions in the profile set, which allows us to visualize the proportion of grid points with poor stability for each layer in the time series of the average profile (Fig. 4b). This visualization takes the concept from Fig. 3e to a temporal context and makes it effortless for users to understand temporal trends in the layerwise stability predictions of *all* profiles within the entire data set within a single, very familiar visualization.

**2.6 Details of snowpack model**

Referee Comment: *Can you please give some more insights of the model behind the modelled snow stratigraphy data? Are you using SNOWPACK or Crocus?*

**Author Response:** We use a weather and snowpack model chain. Our weather model HRDPS (Milbrandt et al. 2016) has a 2.5 km resolution and provides the meteorological forcing for the model SNOWPACK (Bartelt et al. 2002; Lehning et al. 2002b,a).

We added the following new paragraph (P4 L140):

"In this section we present several application examples to illustrate the capabilities of our algorithm. While the snow profile data set used in these examples was simulated with the Canadian weather and snowpack model chain (Morin et al. 2020), our tool can be applied to any simulated snow profile irrespective of its source model. Furthermore, it is possible to use our algorithm on manual profiles, but the processing of these data sets has some unique challenges (see limitation section for more details)."

**2.7 Clarify capabilities in wet snow conditions**

Referee Comment: *The algorithm seems to work dry snow conditions only? Can you comment on that?*

**Author Response:** No, it works equally well for wet snow and melting conditions in spring, see our comment 1.3 and the newly added Fig. 5.

---

## Author Response (AR2)

**Discussion of "A data exploration tool for averaging and accessing large data sets of snow stratigraphy profiles useful for avalanche forecasting"**

**AUTHOR RESPONSE 3**

**Herla et al.**

**July 18, 2022**

Dear editor, dear referees,

We thank both referees for their supportive assessments and their recommendation to publish the manuscript.

We also thank the editor for guiding us through the review process and for the final decision.

In the final revised manuscript we have addressed the small technical corrections/typos pointed out by referee 1. We respond to referee 2 in a point-by-point manner below.

**Responses to Referee #2 (Christoph Mitterer)**

**2.1 Mention flat-field setup**

Referee Comment: *You could state at some point that your simulations are based on flat-field simulations only.*

**Author Response:** Agreed. We added this information in the final manuscript in L125.

**2.2 To scale or not to scale**

Referee Comment: *If I fully understood your approach, you stated that there are now two possibilities within the merging algorithm to handle different snow heights: Either with or without scaling. I was just wondering if a scaled approach would influence the distributions for stability indices that are closely related to depth (e.g. SK38)*

**Author Response:** The two user options of scaling or not scaling the snow profiles are settings of the layer matching algorithm. As long as the layers are matched correctly, the result (of both matching and averaging algorithms) will not be influenced by this setting. While there might specific data sets that call for either one or the other setting, our presented data set produces the same results with the two settings.

**2.3 Details on melt season performance**

Referee Comment: *I know that it is beyond of the scope of your manuscript, but it would be beneficial to add some details within the newly added section on the performance of the algorithm*

*during melt season conditions in spring:*

*> It would be again nice to state that your results are for flat field conditions only and that especially under melting conditions slope and aspect may have considerable impact*

*> I know it is a lot of work, but it would have been cool to see the advantages of the algorithm applied to the approach by Wever et al (2018). You could use the fact that you are not able at the moment to simulate different aspects and it is not feasible at the moment to include the needed RicheardsâTM Equation based routing routine for your grid as arguments for not showing this within the manuscript.*

**Author Response:** We added the following statement in the final manuscript (L225): *"While that data set is suited to highlight how our approach can add value to wet avalanche forecasting, operational simulations must consider slope and aspect processes due to their considerable impact on the melting itself."*

Reproducing the simulation design by Wever et al (2018) is indeed beyond the scope of this manuscript. If this is of interest, it could be a neat case for an ISSW conference paper. In that case you could contribute a data set of regionally distributed snowpack simulations tuned for melting conditions and I will be happy to apply the aggregating post-processing.